# GATA3 induces mitochondrial biogenesis in primary human CD4+ T cells during DNA damage

Lauren A. Callender[1,4,6], Johannes Schroth[1,6], Elizabeth C. Carroll[1,5], Conor Garrod-Ketchley [1], Lisa E. L. Romano[1], Eleanor Hendy[2], Audrey Kelly[2], Paul Lavender[2], Arne N. Akbar [3], J. Paul Chapple[1] & Sian M. Henson [1✉]

GATA3 is as a lineage-specific transcription factor that drives the differentiation of CD4+ T helper 2 (Th2) cells, but is also involved in a variety of processes such as immune regulation, proliferation and maintenance in other T cell and non-T cell lineages. Here we show a mechanism utilised by CD4+ T cells to increase mitochondrial mass in response to DNA damage through the actions of GATA3 and AMPK. Activated AMPK increases expression of PPARG coactivator 1 alpha (PPARGC1A or PGC1α protein) at the level of transcription and GATA3 at the level of translation, while DNA damage enhances expression of nuclear factor erythroid 2-related factor 2 (NFE2L2 or NRF2). PGC1α, GATA3 and NRF2 complex together with the ATR to promote mitochondrial biogenesis. These findings extend the pleotropic interactions of GATA3 and highlight the potential for GATA3-targeted cell manipulation for intervention in CD4+ T cell viability and function after DNA damage.

[1] William Harvey Research Institute, Barts and The London School of Medicine and Dentistry, Queen Mary University of London, London, UK. [2] Peter Gorer Department of Immunobiology and Asthma UK Centre in Allergic Mechanisms of Asthma, King's College London, London, UK. [3] Division of Infection and Immunity, Department of Immunology, University College London, London, UK. [4] Present address: Translational Science, Achilles Therapeutics Ltd, Stevenage Bioscience Catalyst, Stevenage, UK. [5] Present address: Department of Life Sciences, Institute of Technology Sligo, Sligo, Ireland. [6] These authors contributed equally: Lauren A. Callender, Johannes Schroth. ✉email: s.henson@qmul.ac.uk

The function of GATA binding protein 3 (GATA3) in the development of T cells has been well characterised. It is the only member of the GATA family expressed in the T cell lineage[1], and during early T cell development in the thymus GATA3 is essential for CD4+ lineage commitment[2]. Among mature, peripheral CD4+ T cells, GATA3 is also required for CD4+ Type 2 helper (Th2) differentiation[3,4]. More recently, GATA3 has been shown to have many functional roles beyond simply controlling CD4+ Th2 cell differentiation. These roles include the development of invariant NKT cells[5], the maturation and homing of natural killer (NK) cells[6] and the regulation and activation of CD8+ T cells[7]. Collectively these studies expand the role of GATA3 in non-T cell and CD8+ T cell lineages, however the role of GATA3 in non-Th2 CD4+ T cells is unknown.

Here, we show that GATA3 can modulate mitochondrial biogenesis, cell metabolism, as well as antioxidant response regulating proteins via a nucleus to mitochondria signalling axis. GATA3 and AMP-activated protein kinase (AMPK) are activated by a DNA damage response whereby pAMPK increases the transcription of peroxisome-proliferator-activated receptor γ coactivator-1α (PGC1α) and the translation of GATA3. PGC1α and GATA3 then form a complex with the Serine/threonine-protein kinase ATR (ATR) and nuclear factor erythroid 2-related factor 2 (NRF2) to enhance mitochondrial biogenesis to maintain the viability of CD4+ T cells during DNA damage.

## Results

**High levels of GATA3 correlate with increased mitochondrial mass.** As CD4+ T cells become more highly differentiated they develop a Th1-like phenotype, characterised by increased IFN-γ and decreased IL-4, IL-5 and IL-13 production[8,9]. Here, we examined the expression of GATA3 in primary human CD4+ T cell subsets defined by CD45RA and CD27 expression (Supplementary Fig. 1a)—Naïve (CD45RA+CD27+), central memory (CM; CD45RA−CD27+), effector memory (EM; CD45RA−CD27−) and effector memory that re-express CD45RA (EMRA; CD45RA+CD27−)[10,11]. CD4+ EMRA T cells are a highly differentiated population that exhibit many characteristics of cellular senescence such as DNA damage[12], and have a Th1-like phenotype[8,9]. Despite this, unstimulated CD4+ EMRA T cells expressed the highest levels of GATA3 when compared to the other three subsets by flow cytometry and qPCR (Fig. 1a and Supplementary Fig. 1b). Furthermore, the expression of GATA3 remained high in the CD4+ EMRA subset following stimulation (Supplementary Fig. 1c). GATA3 also increased specifically in CD4+CD28− Th1 subsets, defined using CCR4, CCR6 and CXCR3 (Fig. 1b and Supplementary Fig. 1d, e), following the addition of hydroxyurea, which induces DNA damage via double strand breaks[13,14]. Taken together these observations led us to hypothesise that GATA3 has an alternate function in the CD4+ EMRA subset. In the CD8+ lineage, GATA3 expression was low and remained unchanged amongst the four subsets (Supplementary Fig. 2a).

Cellular metabolism is a crucial regulator of T cell function and fate, and the physiological importance of mitochondria in regard to cell metabolism has been widely appreciated for many years. It has been previously shown mitochondrial mass to be lower in CD8+ EMRAs compared to other memory subsets and the mitochondria present to be dysfunctional[15]. Therefore, we sought to investigate the mitochondrial properties of CD4+ T cells subsets. First, we assessed the total mitochondrial mass of CD4+ T cell subsets using the mitochondrial probe; Mitotracker Green. These data revealed that mitochondrial mass increased with differentiation in unstimulated and stimulated CD4+ T cells, with CD4+ EMRA T cells displaying the highest mitochondrial

content in both cases (Fig. 1c and Supplementary Fig. 2b). Additionally, Th1 CD4+CD28− T cells also displayed an increase in mitochondrial mass following the induction of DNA damage (Fig. 1d). Interestingly these changes were only observed in CD4+ CD28− T cells (Supplementary Fig. 1f). Together with increased mitochondrial mass we also observed increased levels of the co-transcriptional regulator important for mitochondrial biogenesis; PGC1α, in CD4+ EMRA T cells by flow cytometry and qPCR (Fig. 1g and Supplementary Fig. 2c, d). Collectively these data positively correlated with the GATA3 expression data and suggested a possible link between GATA3 and mitochondrial biogenesis.

To find out whether the increased mitochondrial mass was of functional importance we used tetramethylrhodamine (TMRE) to measure mitochondrial membrane potential (MMP) (ΔΨm) in CD4+ T cell subsets. Notably, we observed that a greater proportion of CD4+ EMRA T cells had hyperpolarised mitochondria compared with the other three subsets (Fig. 2a). Hyperpolarised mitochondria have been shown to be a source of reactive oxygen species (ROS) in both macrophages[16] and CD8+ tumour infiltrating lymphocytes[17]. Therefore, we examined intracellular ROS and found that CD4+ EMRA T cells also displayed high levels of ROS (Fig. 2b). However, although the levels of ROS were significantly higher in the CD4+ EMRA T cells compared with all other subsets, the ROS/mitochondria ratio was unchanged (Fig. 2b). This led us to postulate that increased mitochondrial mass in CD4+ EMRA T cells is a protective mechanism to ensure that CD4+ EMRA T cells are able to adequately buffer the excess ROS and prevent further DNA damage.

**Senescent CD4+ T cells maintain metabolism.** We next investigated whether increased mitochondrial mass and hyperpolarised MMP in CD4+ EMRA T cells led to metabolic reprogramming. To do this we used a mitochondrial stress test to measure the bioenergetics profiles of TCR stimulated CD45RA/ CD27 defined CD4+ T cell subsets (Fig. 2c). The mitochondrial stress test includes the addition of four pharmacological drugs that manipulate oxidative phosphorylation (OXPHOS), which were added in succession to alter the bioenergetics profiles of the mitochondria. Upon stimulation, naïve CD4+ T cells had a significantly lower basal oxygen consumption rate (OCR) compared to the memory T cells, however no differences were found amongst the three memory subsets (Fig. 2d). The extracellular acidification rate (ECAR), a marker of lactic acid production and glycolysis, was highest in the EM and EMRA CD4+ subsets compared to naïve and CM subsets (Fig. 2e). Furthermore, the spare respiratory capacity (SRC), a measure of how well mitochondria are potentially able to produce energy under conditions of stress, was also significantly lower in the naïve CD4+ T cells but unchanged in the three memory subsets (Fig. 2f). CD4+ T cells remained metabolically quiescent with no stimulation (Supplementary Fig. 2e). Collectively, these data suggest that despite having hyperpolarised mitochondria CD4+ EMRA T cells are able to maintain normal T cell metabolism.

**DNA damage recruits a GATA3 complex inducing mitochondrial biogenesis.** DNA damage can severely impair the function of DNA and plays a major role in age-related diseases and cancer[18]. DNA damage can occur for a number of reasons. For instance, cells which undergo rapid proliferation tend to accumulate DNA damage, as well as many end-stage cells that have undergone replicative senescence in response to telomere dysfunction[19–21]. In support of this we found that DNA damage, determined by an increase in the histone marker for double

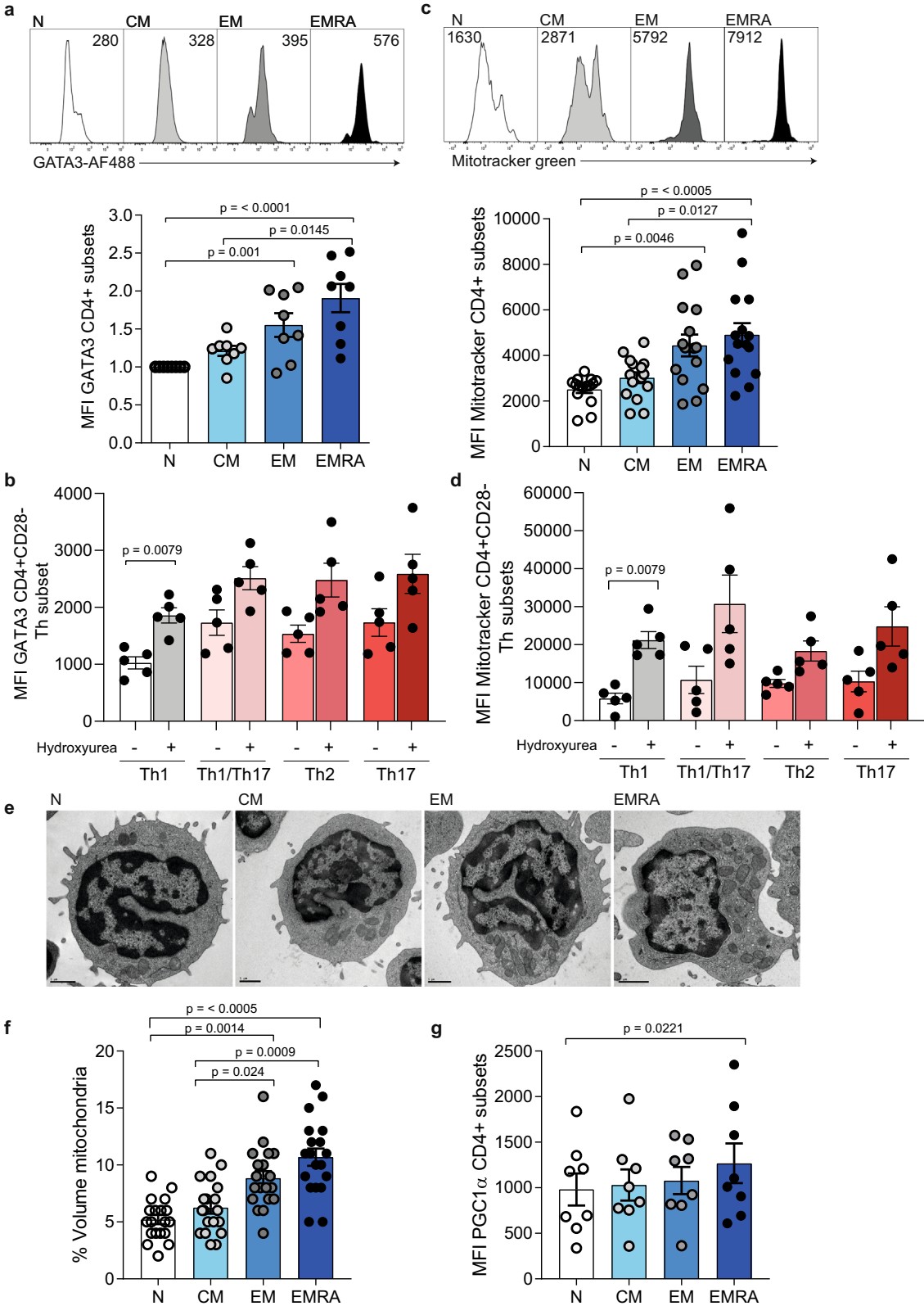

stranded DNA breaks; γH2AX, was significantly higher in the CD4$^+$ EMRA T cells when compared with the other three subsets, and this upregulation correlated to an increased expression of p-p53, a downstream regulator of the DNA damage response (Fig. 3a). In addition, CD4$^+$ EMRA T cells appear to accumulate DNA damage in response to ROS faster than the other memory subsets (Fig. 3b).

A possible connection between GATA3 and mitochondria has been previously reported. A yeast two-hybrid screen showed that GATA3 is able to interact with the DDR proteins; ataxia telangiectasia mutated (ATM) and ataxia telangiectasia and Rad3-related (ATR) as well as PGC1α[22]. Immunoprecipitation (IP) assays showed that GATA3 and PGC1α were bound to the ATR in CD4$^+$ T cells (Fig. 3c). Furthermore, CD4$^+$ EMRA T cells

**Fig. 1 Senescent CD4$^+$ EMRA T cells express high levels of GATA3, which correlates with increased mitochondrial mass. a** Representative flow cytometry histograms and cumulative graph of GATA3 staining in CD27/CD45RA defined CD4$^+$ T cell subsets ($n = 8$ biologically independent samples). **b** Graphs showing GATA3 MFI in CD4$^+$CD28$^-$ Th subsets determined using CCR4, CCR6 and CXCR3 ($n = 5$ biologically independent samples). **c** Representative flow cytometry histograms and cumulative graph of Mitotraker Green staining in CD27/CD45RA defined CD4$^+$ T-cell subsets ($n = 14$ biologically independent samples). **d** Graphs showing Mitotracker Green staining in CD4$^+$CD28$^-$ Th subsets determined using CCR4, CCR6 and CXCR3 ($n = 5$ biologically independent samples). **e** Electron microscope images of CD27/CD45RA defined CD4$^+$ T cell subsets imaged directly ex vivo. Scale bars 1 µM. Representative for 20 different images. **f** Graph shows the percentage by cell volume of mitochondria in CD27/CD45RA defined CD4$^+$ T cell subsets determined by a point-counting grid method from 20 different electron microscope images from three individuals. **g** PGC1α staining in CD45RA/CD27 defined CD4$^+$ T cell subsets ($n = 8$ biologically independent samples). $p$ values were calculated using a two-way Friedman test followed by Dunn multiple comparison for post hoc testing for parts **a**, **c**, **f** and **g** and a two-way Mann–Whitney $U$ test for parts **b** and **d**. Graphs show ± SEM.

had the highest levels of GATA3 and PGC1α bound to the ATR in all three repeats (Supplementary Fig. 3a). We next sought to identify other mitochondrial regulators that may also be upregulated by GATA3 using chromatin immunocleavage on resting naïve CD4 T cells. Despite GATA3 being preferentially expressed by memory cells, these data revealed that GATA3 may be recruited directly or indirectly to the promoter regions of a number of genes including the *NRF1* and *NFE2L2/NRF2* respectively, *SOD3*, as well as *PPARGC1A* (Fig. 3d). DNA damage, induced by overnight incubation with hydroxyurea, resulted in a significant increase in Tom20, a major outer mitochondrial membrane receptor and therefore mitochondrial marker, and GATA3 using confocal analysis. The increased expression of GATA3 was located in both the cytoplasm and nucleus (Fig. 3e and Supplementary Fig. 3b). In addition, flow cytometry revealed a significant increase in phospho-p53, intracellular PGC1α and an increased mitochondrial content, determined by elevated Mitotracker green staining (Supplementary Fig. 3c–e). Furthermore, IP assays showed that treatment with hydroxyurea increased the amount of PGC1α, NRF2 and SOD3 bound to GATA3 (Fig. 3f and Supplementary Fig. 3f). Interestingly we found no interaction between NRF1 and GATA3, suggesting that GATA3 may regulate NRF1 at the level of transcription rather than via protein–protein interactions.

In order to determine the requirement for ATR activation in regulating mitochondrial biogenesis we used the ATR inhibitor AZD6738 (Ceralasertib). We found that inhibition of the ATR following hydroxyurea treatment prevented the increase in both GATA3 expression and mitochondrial biogenesis, bringing GATA3 and mitochondrial mass down to the level observed with stimulation alone (Fig. 3g). We then sought to determine the confirmation of the complex through siRNA knockdown of GATA3 followed by pull down of PGC1α. We found ATR present in both the control and GATA3 siRNA lanes suggesting that while ATR activity is required for complex formation, it may not exclusively bind to PGC1α in the presence of GATA3.

**Metabolism and mitochondrial biogenesis regulated by GATA3 and AMPK.** To verify that GATA3 can directly regulate mitochondrial biogenesis and fitness we used small interfering RNA (siRNA) to reduce GATA3 expression in purified human CD4$^+$ T cells following activation (Supplementary Fig. 4a–c). Transfection with GATA3 siRNA led to a significant decrease in both the levels of PGC1α and Mitotracker Green staining (Fig. 4a, b and Supplementary Fig. 4d), signifying a marked reduction in mitochondrial mass. Confocal microscopy examination of Tom20 (Fig. 4c) showed a significant reduction in both mitochondrial mass (Supplementary Fig. 4e) and volume (Supplementary Fig. 4f) in Jurkat T cells following transfection with GATA3-siRNA. TMRE staining revealed a dramatic shift from hyperpolarised to hypopolarised MMP in the GATA3 siRNA transfected CD4$^+$ T cells but not the scrambled siRNA transfected CD4$^+$ T cells (Fig. 4d). Moreover, transfection with GATA3 siRNA

dramatically inhibited the ability of CD4$^+$ T cells to perform OXPHOS when compared with CD4$^+$ T cells transfected with negative control scrambled siRNA (Fig. 4e). Basal OCR and SRC were significantly decreased in the GATA3 siRNA transfected CD4$^+$ T cells (Supplementary Fig. 4g). CD8$^+$ T cells transfected with GATA3 siRNA showed no difference in their bioenergetic profiles when compared with negative control scrambled siRNA (Supplementary Fig. 4h). Furthermore, we failed to observe a switch to glycolysis with GATA3 knockdown, as ECAR did not significantly increase with the addition of oligomycin (Fig. 4f). Additionally, chromatin immunocleavage demonstrated that GATA3 also bound to the promoter regions of hexokinase 1 and 2, together with SLC2A1 suggesting that GATA3 knockdown could interfere with glycolytic metabolism (Supplementary Fig. 5a). A finding that is supported by data showing that GATA3 controls glycolysis through the negative regulation of PPARγ[23].

To determine whether GATA3 changed the transcription of the AMPK, PGC1α, SOD3, NRF1 and NRF2 genes we performed further siRNA knockdowns in the presence of hydroxyurea (Supplementary Fig. 5b). Paradoxically knockdown of GATA3 caused increased expression of *PRKAA2* (AMPK) and *PPARGC1A* (PGC1α) mRNA and a reduction in the transcription of *SOD3* and *NRF1* and *NFE2L2* (NRF2). Increased PGC1α transcription despite downstream signalling impairment has been observed previously and was associated with an energetic stress response[24]. Our data would suggest that GATA3 controls the transcription of key redox sensing components and that its loss leads to metabolic stress[25].

We then sought to measure how GATA3 co-ordinates mitochondrial biogenesis. We found that pAMPK was highly expressed in the CD4$^+$ EMRA subset ex vivo (Supplementary Fig. 5c) and expression remained high following stimulation (Supplementary Fig. 5d). Furthermore, pAMPK was found to increase when DNA damage was induced following overnight incubation with hydroxyurea (Fig. 4g and Supplementary Fig. 5e). Our finding mirrors a reported role for the ATM in mitochondrial biogenesis through AMPK activation in response to DNA damage[26]. However, while chromatin immunocleavage revealed that GATA3 can be recruited to the promoter regions of both AMPK1 and 2 (Supplementary Fig. 5f), we found no interaction between GATA3 and AMPK by IP assays (Supplementary Fig. 5f). This led us to believe that GATA3 does not regulate AMPK through protein–protein interactions. We then assessed gene expression of GATA3, PGC1α and NRF2 by qPCR following hydroxyurea treatment in Jurkat T cells. We found only the transcriptional activity of PGC1α to be upregulated by DNA damage (Fig. 4h). This increase in PGC1α mRNA was largely controlled by AMPK, for the addition of an AMPK inhibitor, Compound C (Dorsomorphin), significantly reduced mRNA levels (Fig. 4i). It has been reported that TCR signalling can increase GATA3 protein expression not by increasing its mRNA level but rather by enhancing the rate of translation[27]. We therefore tested whether AMPK controlled the amount of

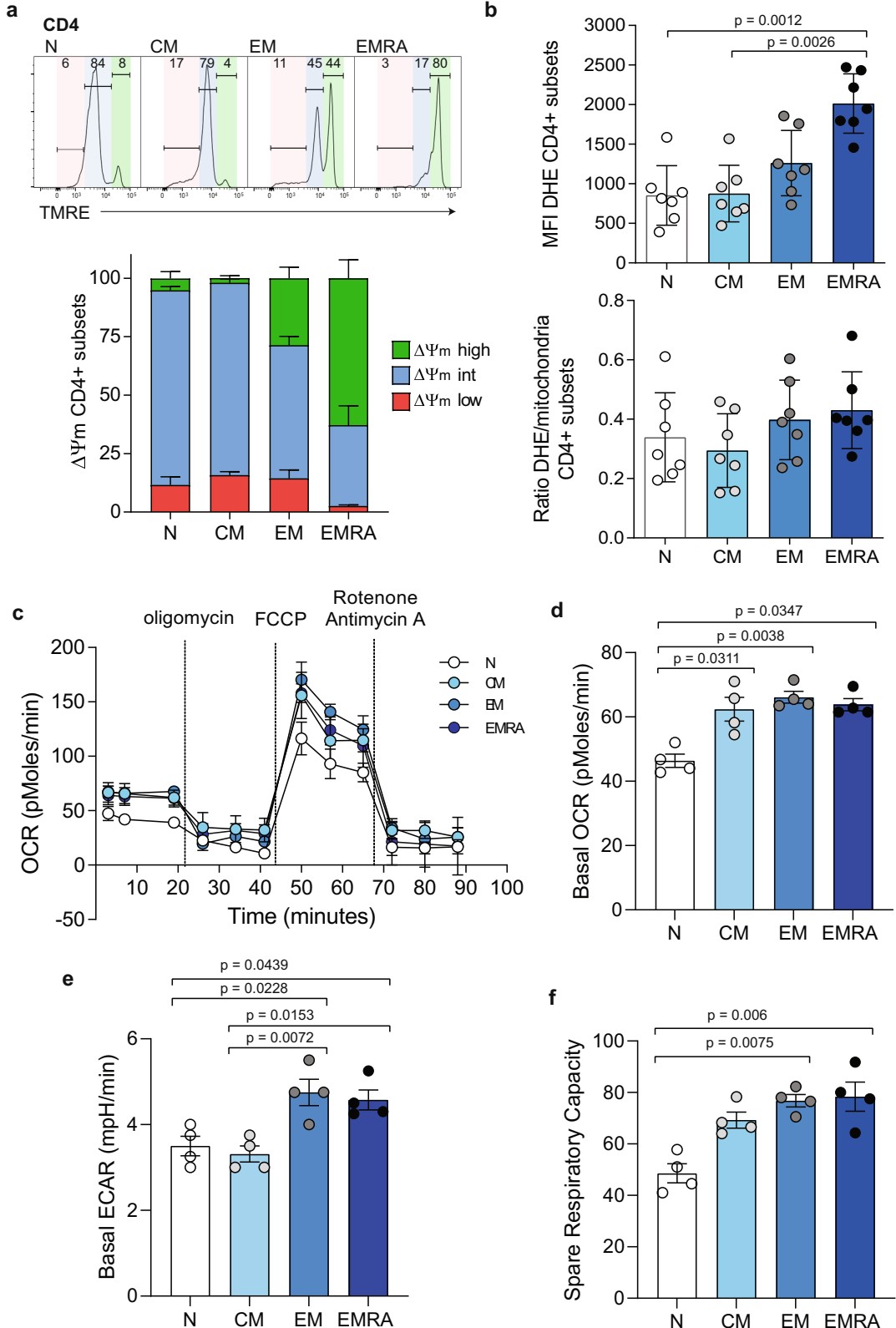

**Fig. 2 CD4+ EMRA T cells have hyperpolarised mitochondria, but maintain normal T-cell metabolism. a** Mitochondrial membrane potential (MMP) was determined in CD27/CD45RA defined CD4+ T cell subsets using TMRE ($n = 5$ biologically independent samples). Low MMP = red, intermediate MMP = blue, high MMP = green. **b** ROS production was obtained using DHE in the CD27/CD45RA defined CD4+ T cell subsets ($n = 7$). **c** OCR of the CD4+ T cell subsets was measured after 15 min stimulation with 0.5 μg/ml anti-CD3 and 5 ng/ml IL-2, cells were subjected to a mitochondrial stress test using indicated mitochondrial inhibitors. Data are representative of four independent experiments. **d** Basal OCR, **e** Basal ECAR and **f** the SRC of the CD4+ T cell subsets after 15 min stimulation with 0.5 μg/ml anti-CD3 and 5 ng/ml IL-2 ($n = 4$ biologically independent samples). *p* values were calculated using a two-way Kruskal–Wallis test followed by Dunn multiple comparison for post hoc testing. Graphs show ± SEM.

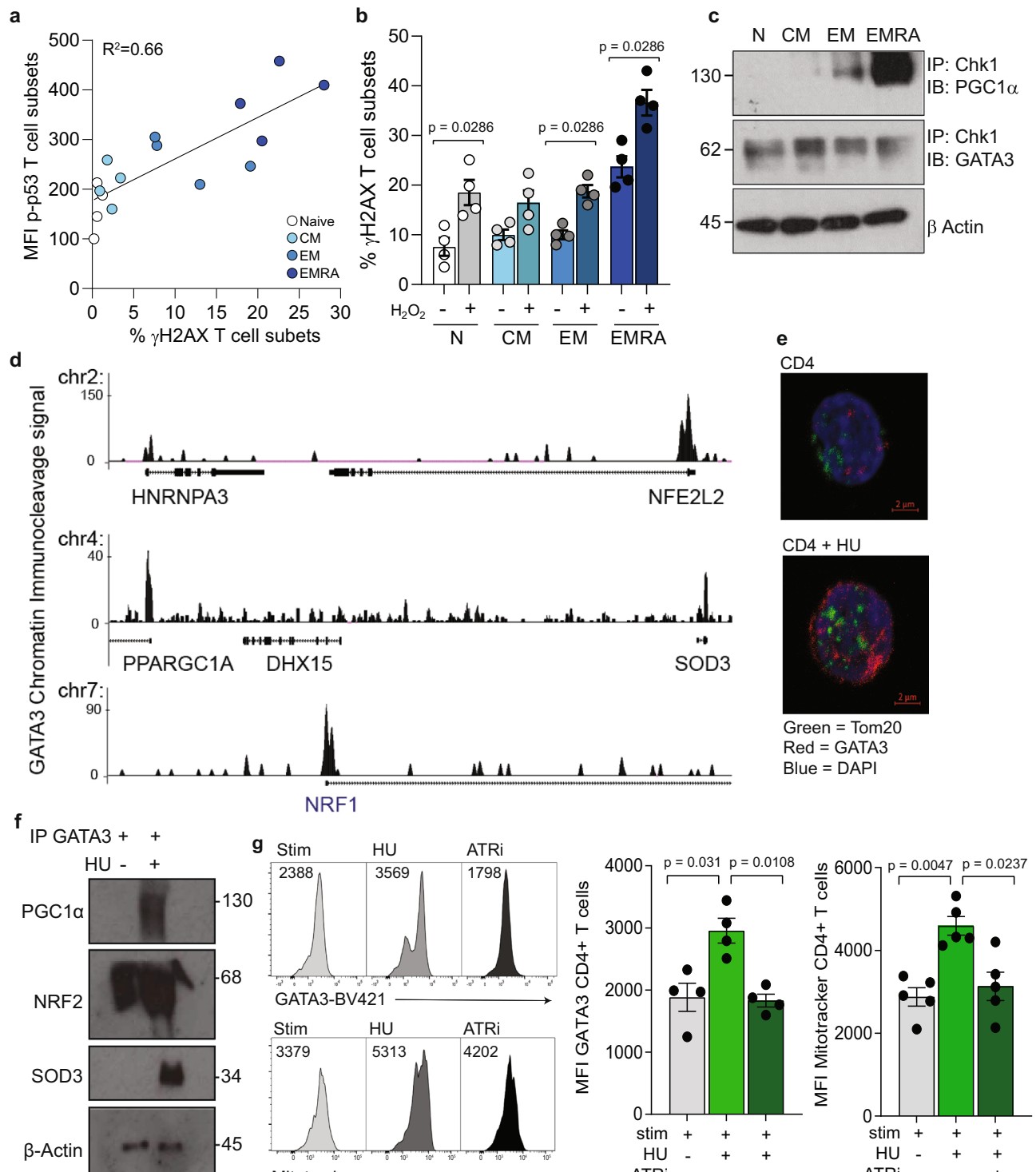

**Fig. 3 DNA damage recruits a GATA3-PGC1α complex to induce mitochondria biogenesis in CD4+ EMRA T cells. a** Linear regression analysis of the DNA damage marker γH2AX and p-p53 in CD27/CD45RA defined CD4+ T cell subsets ($n = 4$ biologically independent samples). **b** DNA damage in CD4+ T-cell subsets following the induction of oxidative stress using $H_2O_2$ ($n = 4$ biologically independent samples). **c** Western blot showing Chk1 immunoprecipitation in the CD27/CD45RA defined CD4+ T cell subsets quantifying the presence of GATA3 and PGC1α bound to the Chk1 antibody ($n = 3$ biologically independent samples). **d** Chromatin immunocleavage showing GATA3 binding to the promoter regions of NRF1 (*NRF1*), NRF2 (*NFE2L2*), SOD3 (*SOD3*), and PGC1α (*PPARGC1A*) in naïve CD4+ T cells. **e** Tom20 and GATA3 staining in CD4+ T cells following treatment with 400 μM hydroxyurea ($n = 3$ biologically independent samples). **f** Western blot showing GATA3 immunoprecipitation in CD4+ T cells following hydroxyurea treatment quantifying the presence of PGC1α, NRF2 and SOD3 ($n = 3$ biologically independent samples). **g** Representative flow cytometry histograms and graph of GATA3 and mitochondrial mass measured using mitotracker green following treatment of CD4+ T cells with and without hydroxyurea and 1 μM AZD6738 ($n = 4$ biologically independent samples). $p$ values were calculated using a two-way paired Mann–Whitney $U$ test for part **b** and a Kruskal–Wallis test followed by Dunn multiple comparison for post hoc testing for part **g**. Graphs show SEM.

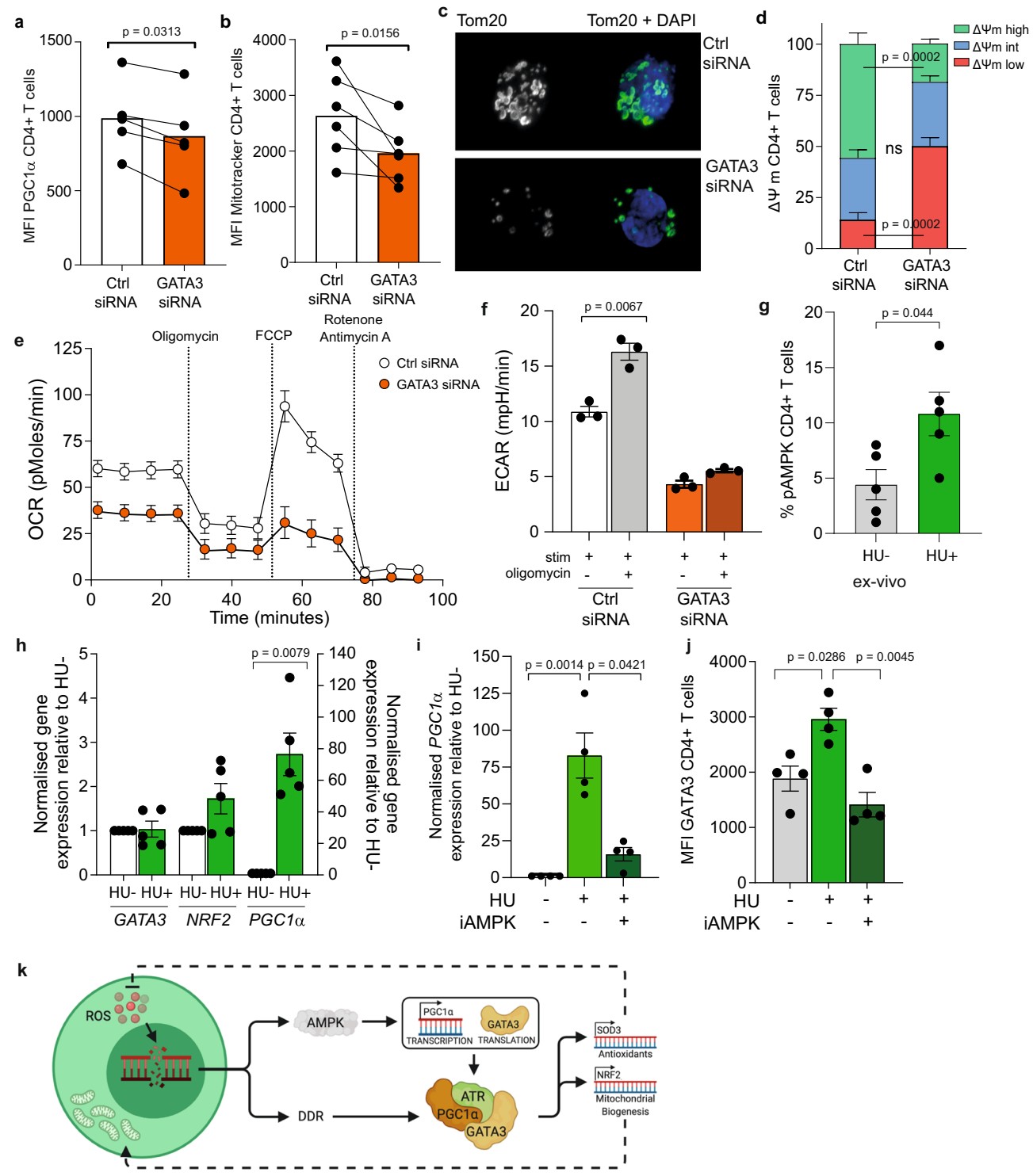

GATA3 protein and found that overnight incubation with Compound C, an AMPK inhibitor, caused a significant loss of GATA3 protein in CD4+ T cells (Fig. 4j and Supplementary Fig. 6a). Whilst we saw no increase in the transcription of *NFE2L2* (NRF2) following DNA damage we did find increased protein expression (Supplementary Fig. 6b). The translocation of NRF2 to the nucleus has been shown to be regulated by the activity of AMPK[28], however we found no change in location of NRF2 following treatment with the AMPK inhibitor (Supplementary Fig. 6c), suggesting that NRF2 is regulated by the DDR but not AMPK.

While c-myc has been shown to play a role in B cell mitochondrial biogenesis[29], we find no evidence for its involvement in this model. Despite chromatin immunocleavage showing GATA3's recruitment to the promoter region of c-myc (Supplementary Fig. 6d) we do not see any change in c-myc expression post-transcriptionally following knockdown of GATA3 (Supplementary Fig. 6e). Mitochondrial biogenesis mediated via c-myc is thought to be regulated by NRF1 gene targets[30], as we failed to find evidence of NRF1 involvement we believe that NRF1-c-myc do not control mitochondrial biogenesis through DNA damage.

**Fig. 4 GATA3 and AMPK regulate CD4+ T-cell metabolism and mitochondrial biogenesis.** Graphs showing (**a**) PGC1α and (**b**) Mitotracker Green staining in whole CD4+ T cells transfected with either control or GATA3 siRNA-AF647 (n = 5 and 6, respectively, biologically independent samples). p values were determined using a one-way Wilcoxon matched-pairs signed rank test. **c** Tom20 confocal staining in Jurkat T cells transfected with either control or GATA3 siRNA-AF647, representative images of three independent experiments. **d** MMP was determined in whole CD4+ T cells transfected with either control or GATA3 siRNA-AF647 using TMRE (n = 6 biologically independent samples). Low MMP = red, intermediate MMP = blue, high MMP = green. p values were determined using a two-way ANOVA. **e** OCR of whole CD4+ T cells transfected with either control or GATA3 siRNA-AF647 was measured after 15 min stimulation with 0.5 μg/ml anti-CD3 and 5 ng/ml IL-2, cells were subjected to a mitochondrial stress test using indicated mitochondrial inhibitors. Data representative of three biologically independent samples. **f** ECAR responses following treatment with oligomycin of the CD4+ siRNA transfected cells stimulated as described above (n = 3 biologically independent samples). p values were determined using a two-way Mann–Whitney U test. **g** Staining of pAMPK (Thr172) in CD4+ T cells following overnight incubation with 400 μM hydroxyurea (n = 5 biologically independent samples). p values were determined using a two-way Mann–Whitney U test. **h** qPCR data using Jurkat T cells following mRNA levels of GATA3, PGC1α and NRF2 following overnight incubation with hydroxyurea (n = 5 biologically independent samples). All results are shown as ΔΔCT normalised against untreated samples. p values were calculated using a two-way Mann–Whitney U test. **i** qPCR data of PGC1α following treatment of Jurkat T cells with and without hydroxyurea and 10 μM Compound C (n = 4 biologically independent samples). p values were calculated using a Kruskal–Wallis test followed by Dunn multiple comparison for post hoc testing. **j** Staining of GATA3 following treatment of CD4+ T cells with and without hydroxyurea and 10 μM Compound C (n = 4 biologically independent samples). p values were calculated using a Kruskal–Wallis test followed by Dunn multiple comparison for post hoc testing. **k** Potential mechanism of GATA3 and AMPK leading to mitochondrial biogenesis during DNA damage. Graphs show ± SEM.

Finally, we sought to investigate the functional consequence of preventing increased mitochondrial mass in response to DNA damage. We prevented mitochondrial biogenesis using doxycycline, an antibiotic that inhibits mitochondrial protein translation[31], as the use of PGC1α siRNA was lethal to CD4+ T cell. When doxycycline was given following hydroxyurea treatment, we observed a reduction in mitochondrial content as well as a dramatic increase in ROS production (Supplementary Fig. 6f, g). Moreover, the decline in mitochondrial content led to a rise in apoptosis, evident by the sharp increase in annexin expression (Supplementary Fig. 6h). Doxycycline treatment resulted in a CD4+ T cell that resembled a senescent CD8+ T cell[32]. Therefore, the presence of GATA3 in CD4+ T cells has the potential to slow the acquisition of senescence.

Taken together, these data provide evidence that GATA3 modulates mitochondrial biogenesis, cell metabolism, as well as antioxidant response regulating proteins via a nucleus to mitochondria signalling axis to maintain the viability of these cells during DNA damage.

## Discussion

Mechanistically, GATA3 and AMPK are activated by a DNA damage response; pAMPK increases expression of PGC1α at the level of transcription and GATA3 at the level of translation. PGC1α and GATA3 complex together with the ATR and NRF2 to enhanced mitochondrial biogenesis.

Evidence for GATA3 being a metabolic regulator is growing. Son et al. recently demonstrated that GATA3 directly binds to PGC1α to regulate thermogenesis in white adipose tissue via increased UCP-1 expression[33]. We demonstrate here that in CD4+ T cells, PGC1α is activated by both GATA3 and AMPK. DNA damage acting through the ATM has been shown to activate AMPK in numerous tissues and can directly interact and phosphorylate PGC1 increasing its transcriptional activity[34], as seen here. However, we do not find AMPK to regulate the expression of NRF2 despite it being shown that the translocation of NRF2 to the nucleus is regulated by the activity of AMPK. AMPK phosphorylates NRF2 at Ser550, which results in GSK3β inactivation, both of which are essential for NRF2 translocation to the nucleus[28]. However, direct interaction between PGC1α and NRF2 has been demonstrated where p38 inactivates GSK3β[35]. As p38 activity is high in end-stage CD4+ T cells[12] we cannot discount a role for p38 in this model. GATA3 has also been shown to control metabolic reprograming of CD8+ T cells following stimulation via the T cell receptor. Knockdown of GATA3

reduced both glycolysis and OXPHOS via c-myc[36]. While we do not find evidence for the involvement of c-myc in our complex it does add weight to GATA3 being a metabolic regulator acting via different mechanisms dependent on cell type and method of activation.

Importantly, as DDR mechanisms are widely conserved mechanisms, it is likely for this mechanism to not be restricted to the lymphocyte lineage alone. GATA3 is expressed in various cell types outside the haematopoietic system and has been implicated in the tumorigenesis of many cancers, such as luminal breast cancer[37], neuroblastoma[38] and endometrial carcinomas[39].

GATA3-positive breast cancers have been shown to be highly differentiated, with resulting tumours two-fold larger in size than control tumours. In contrast, GATA3-negative breast cancers form poorly differentiated tumours and are highly metastatic[40]. However, despite being associated with low metastatic potential and therefore a more favourable prognosis, GATA3-positive tumours are also associated with resistance to chemotherapy[41]. In light of the present study, we hypothesise that the accumulation of DNA damage generated during chemotherapy recruits GATA3, allowing GATA3-positive tumours to maintain mitochondrial fitness and evade chemotherapy induced apoptosis. However, additional examination of GATA3 and its contribution to mitochondrial biogenesis in cancer is needed to confirm this.

More broadly, our results implicate that transcription factors are likely to serve additional functions after their initial roles in differentiation. This is already the case for the Th1 differentiation marker; T-bet, which is also known to be crucial for long-term memory in CD8+ T cells[42,43] and maturation of NK cells[44]. More recently; STAT5, which has traditionally been associated with cytokine signalling and JAK kinases[45], has now also been identified as a central regulator of naïve CD4+ T cell metabolism[46]. Furthermore, although the GATA proteins have restricted expression patterns to some extent their functions can be interchangeable. For example, GATA1, -2, and -4 can all activate the expression of GATA3 target genes IL-4 and IL-5 and repress IFNγ in T cells[47]. Interestingly, GATA4 has previously been shown to become activated in the presence of the ATM and ATR, resulting in the induction of senescence and a senescence associated secretory phenotype in human fibroblasts[48]. These functional overlaps suggest that other GATA proteins may also be capable of influencing cell metabolism.

In summary, our data identify a role for GATA3 in response to DNA damage, revealing it as a key mediator of mitochondrial biogenesis and cell metabolism. Our findings extend the

pleotropic nature of GATA3, demonstrating that more focus should be placed on investigating the role of other transcription factors in later stages of cell development to assess the full range of their abilities. Furthermore, the widespread expression of GATA3 in numerous cancers demonstrate the potential for GATA3-targeted manipulation for clinical interventions.

## Methods

**Blood sample collection and isolation.** Ethics for blood collection were approved by the NRES Committee North East (REC reference: 16/NE/0073) and all donors provided written informed consent. Heparinized peripheral blood samples were taken from healthy volunteers, average age 41 years ±5. Healthy volunteers had not had an infection or immunisation within the last month, no known immunodeficiency, and had not received any other immunosuppressive medications within the last 6 months. Peripheral blood mononuclear cells (PBMCs) were isolated using Ficoll-Paque (Amersham Biosciences).

**Flow cytometric analysis and cell sorting.** Flow cytometric analysis was performed on ~$1 \times 10^6$ PBMCs per sample. Cells were incubated in the dark with the Zombie_NIR live/dead stain (1:1000, Biolegend) and the following antibodies: CD4 PE-CF594 (1:100, RPA-T4) from BD Biosciences, CCR4 PE (1:40, L291H4), CCR6 APC (1:40, G034E3), CXCR3 APC-cy7 (1:40, G025H7), CD45RA BV605 (1:100, HI100) and CD27 BV421 (1:400, O323) from BioLegend, for 15 min at room temperature. For intracellular staining, the following antibodies were used: GATA3 AF488 (1:20, 16E10A23), GATA3 BV421 (1:20, 16E10A23) and γH2AX AF488 (1:20, 2F3) from BioLegend, PGC1α (1:50, 3G6), pAMPK (1:50, Thr172, 40H9), p53 AF647 (1:50, 1C12) and p-p53 AF647 (1:50, Ser15, 16G8) from Cell Signalling Technology, c-myc (1:50, 9E10) from Santa Cruz Biotechnology and goat anti-rabbit IgG H&L Alexa Fluor 488 (1:1000, AF488) from Abcam.

Detection of GATA3, PGC1α, p53, γH2AX, pAMPK and p-p53 were carried out directly ex vivo. The Foxp3 Transcription Factor Staining Buffer Kit (Biolegend, 566349) was used for staining GATA3, p53, γH2AX, pAMPK and p-p53. Following surface staining PBMCs were incubated with the FOXP3 Fix/Perm buffer for 20 min in the dark at room temperature followed by incubation with the FOXP3 Perm buffer (10X) for 15 min in the dark at room temperature. Finally cells were incubated with the FOXP3 Perm buffer (10X) and appropriate amount of the antibody being used for 30 min in the dark and room temperature. The FIX & PERM® Cell Permeablilisation Kit (Life Technologies, GAS003) was used to enable intracellular staining of PGC1α. Immediately after surface staining, PBMCs were fixed with Reagent A for 15 min in the dark at room temperature. Next the cells were permeabilised with Reagent B plus 10% goat serum and recommended volume of primary intracellular antibody. Cells were incubated for 20 min in the dark at room temperature followed by a 20 min incubation with the secondary antibody.

All samples were acquired on a LSR Fortessa (BD Biosciences) equipped with four lasers: 488 nm blue laser, 561 nm yellow green laser, 641 red laser and 405 nm violet laser and data were analysed using FlowJo software.

CD4+ T cells were purified using anti-CD4+ conjugated microbeads (Miltenyi Biotec, 130-045-101) according to the manufacturer's instructions. Positively selected CD4+ T cells were labelled with CD27 FITC (1:20, O323) and CD45RA APC (1:20, HI100) both from BD Biosciences and sorted using a FACSAria (BD Biosciences). The purity of CD4 T-cell subsets were assessed by flow cytometry.

**Mitochondrial measurements.** For mitochondrial mass, PBMCs were incubated with 100 nM MitoTracker Green FM (Thermo Fisher, M7514) for 30 min at 37 ℃, 5% CO₂. ROS was measured using Dihydroethidium (Thermo Fisher, D23107), 5 μM MitoSOX (Thermo Fisher, M36008) was incubated with labelled PBMCs for 20 min at 37oC, 5% CO₂. MMP was investigated using TMRE (Thermo Fisher, T669) 2 μM TMRE was incubated with PBMCs for 30 min at 37 ℃, 5% CO₂. Mitochondrial dyes can alter staining patterns, this was checked using fluorescence minus one. Following mitochondrial staining, samples were surface stained and then left unfixed and immediately analysed on an LSR Fortessa (BD Biosciences).

To assess mitochondrial morphology by confocal microscopy, Jurkat cells were plated onto microscope slides then fixed and permed using the FIX & PERM® Cell Permeablilisation Kit (Life Technologies, GAS003) to enable intracellular labelling. Cells were incubated with Reagent A (Fixation Medium) for 15 min at room temperature, then Reagent B (Permeablisaton Medium) plus 10% goat serum and Tom20 (1:100, F-10, Santa Cruz) for 20 min at room temperature followed incubation with the secondary antibody. Finally, cells were incubated with 4′,6-diamidino-2-phenylindole (Thermo Fisher, 62248) for 10 min at room temperature in the dark, and then fixed in 2% paraformaldehyde. Samples were imaged on a Zeiss LSM 880 confocal microscope with a ×63 oil-immersion objective lens. Excitation was at 488 nm from an argon-ion laser. Fluorescence detection was in the green; 500–570 nm and UV; 405 nm channel. Quantification of image data was performed by measuring the intensity of the 500–570 nm channel using ZEN imaging. Imaris Image Analysis software was used to determine mitochondrial volume through surface 3D rendering of Z-stack images.

**Transmission electron microscopy studies.** CD27/CD45RA defined CD4+ T-cell subsets were isolated and fixed in 2% paraformaldehyde, 1.5% glutaraldehyde in 0.1 m phosphate buffer pH 7.3. They were then osmicated in 1% OsO4 in 0.1 M phosphate buffer, dehydrated in a graded ethanol-water series, cleared in propylene oxide and infiltrated with Araldite resin. Ultra-thin sections were cut using a diamond knife, collected on 300 mesh grids, and stained with uranyl acetate and lead citrate. The cells were viewed in a Jeol 1010 transmission electron microscope (Jeol) and imaged using a Gatan Orius CCD camera (Gatan). Mitochondrial volume density (percentage of T-cell volume occupied by mitochondria) was determined from EM images using a point-counting method using image J.

**Live cell metabolic assays.** The Cell Mito Stress Test was performed on a Seahorse XFe96 Analyser (Agilent) to assess mitochondrial function and OXPHOS. Seahorse sensor cartridges (Agilent, 101085-004) were incubated at 37 ℃ in a non-CO₂ incubator overnight with Seahorse XF Calibrant pH 7.4 (Agilent) to rehydrate the sensor probes. The assay was performed in RPMI-1640 buffer without phenol red and carbonate buffer (Sigma-Aldrich) containing 25 mM glucose, 2 mM L-glutamine and 1 mM pyruvate with a pH of 7.4. The stress test was performed using 1 μM Oligomycin, 1.5 μM flurocarbonyl cyanide phenylydrazone, 100 nM rotenone, and 1 μM antimycin A (Agilent, 103015-100). Cells were stimulated with 1 μg/μl of anti-CD3 and 5 ng/ml IL-2 15 min prior to each metabolic test. Metabolic measurements were calculated according to the manufacturer's guidelines.

**Immunoprecipitation.** For IP experiments the Chk1 (1:2, E250, Abcam) or GATA3 (1:2, 16E10A23, Biolegend) antibody was incubated with Red Protein G Affinity beads (Sigma-Aldrich) for 1 h at 4 ℃ on a shaker to allow the antibody to bind to the beads. Cell lysates were collected in RIPA buffer plus inhibitors (protease: complete mini Roche; 11836153001, and phosphatases: cocktail set II Calbiochem; 524625) and then incubated with the antibody-bead mix overnight at 4 ℃ on a shaker. The following morning, samples were centrifuged, and the supernatant was discarded leaving only the proteins able to bind to Chk1 or GATA3. SDS-polyacrylamide electrophoresis and western blotting of proteins of interest including, GATA3 (1:1000, 16E10A23, Biolegend), PGC1α (1:50, 3G6, Cell Signalling), Nrf1 (2 μg/ml, 492413, R&D Systems), Nrf2 (1 μg/ml, ab89443, Abcam), SOD3 (20 μg/ml, AF3420, R&D Systems), ATR (1:1000, E1S3S, Cell Signalling) and βactin (1:1000, 13E5, Cell Signalling) was then performed. Protein was detected via chemiluminescence by means of the ECL system (GE Healthcare, GERPN2106).

**Hydrogen peroxide experiments.** Oxidative stress was induced by the addition of 200 μM H₂O₂ for 1 h. After this time the expression of γH2AX (1:20, 2F3, Biolegend) in CD27/CD45RA defined CD4+ T-cell subsets was assessed by flow cytometry.

**Hydroxyurea experiments.** To induce DNA damage, PBMCs were incubated with 400 μM hydroxyurea for either 24 or 48 h as specified. Where indicated 1 μM AZD6738, 10 μM Compound C (Dorsomorphin) or 50 μM doxycycline was also added to the cell cultures, DMSO was added to control samples.

**Apoptosis experiments.** PBMCs were incubated with various inhibitors, after which time the cells were stained using the Annexin V apoptosis kit (Biolegend, 640914) according to the manufacturer's instructions.

**qPCR.** RNA from CD27/CD45RA defined CD4+ T-cell subsets or Jurkat T cells were isolated using the RNeasy kit (Qiagen, 74004) according the manufacturer's instructions. Transcripts were quantified using the High-Capacity cDNA Reverse Transcription Kit (Applied Biosystems, 4368814) and the SsoAdvanced Universal SYBR Green Supermix (Bio-Rd Laboratories, 1725271) according the manufacturer's instructions. Primers were purchased from IDT and sequences can be found in Supplementary Table 1.

**GATA3 siRNA knockdown experiments.** GATA3 and scrambled siRNA (GATA3; 5′-/5Alex647N/UAG GCG AAU CAU UUG UUC AAA-3′, Scrambled; 5′ CCU GUU CUU AAA AUA GUA GGC 3′) purchased from IDT were dissolved to a final concentration of 10 μM in electroporation buffer, transferred to an electroporation cuvette and electroporated using programme T-023 on an Amaxa Nucleofector (Lonza Bioscience, VPA-1002). Cell were then left to recover for 24–48 h in fresh RPMI-1640 media. Primary human CD4+ T cells were stimulated with 1 μg/μl of anti-CD3 and 5 ng/ml IL-2 for 24 h prior to transfection whereas Jurkat T cells were left unstimulated.

**Chromatin immunocleavage.** Chromatin immunocleavage was undertaken using protein A-micrococcal nuclease (pA-MN)[49] repurposed as a Cut and Run assay[50]. Naïve primary human T cells were isolated from PBMCs using a CD4 Positive Isolation Kit (Thermo Fisher, 11331D) according to the manufacturer's instructions. Naive CD45RA+ cells were purified from CD4+ cells by depletion of CD45RO+ cells using mouse anti-human CD45RO Ab (1:20, UCHL1; BD

PharMingen) and anti-mouse IgG2a Dynabeads (Thermo Fisher, 11033) according to the manufacturer's instructions. Following nuclear isolation, regions of chromatin bound by GATA3 were identified using a mouse anti-GATA3 antibody (1:100, HG3-31) Santa Cruz Biotechnology followed by pA-MN. DNA was isolated from eluted chromatin, libraries were prepared using NEBNext® Ultra II DNA Library Prep Master Mix Set and Multiplex Oligos for Illumina® (New England Biolabs, E7760). Library quality was assessed using Bioanalyzer 2100 High Sensitivity DNA Gels (Agilent). Libraries were subjected to 50 bp single end read sequencing on an HighSeq 2500 (Illumina) in rapid run format and reads were aligned to Human genome hg19 using Bowtie2 (Galaxy v2.2.6)[51] and visualised within the UCSC genome browser.

**Statistical analysis**. GraphPad Prism was used to perform statistical analysis. Statistical significance was evaluated using a one-way Wilcoxon matched-pairs signed rank test, a two-sided paired Mann–Whitney U test, a two-way ANOVA, a Kruskal–Wallis or a Friedman test followed by Dunn multiple comparison for post hoc testing. The statistical test performed for each experiment is listed in the figure legends. Graphs show SEM. Differences were considered significant when $p$ was <0.05.

**Reporting summary**. Further information on research design is available in the Nature Research Reporting Summary linked to this article.

## Data availability

Chromatin immunocleavge data have been deposited in the GEO database under the primary accession number GSE172346. The data that support the findings of this study are available from the corresponding author upon reasonable request. Source data are provided with this paper.

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

## Acknowledgements

This work was supported by the British Heart Foundation FS/15/69/32043 (LAC), a Springboard award from the Academy of Medical Science and the Wellcome trust SBF001\1013 (ECC, SMH), the Rosetrees Trust M731 (SMH), Barts and the London Charity MGU0536 (CGK) and Royal College of Anaesthetists WRO-2018-0065 (JS). Asthma UK funded the PhD studentship of EH and the BRC at Guy's and St.Thomas' Hospitals funded AK. ANA was funded by the Medical Research Council (MR/P00184X/ 1). The LSM 880 confocal used in these studies was purchased through a Barts and the London Charity grant MGU0293. We would also like to thank Dr. Mark Turmaine for his technical assistance with the electron microscopy; the CRUK Flow Cytometry Core Service at Barts Cancer Institute (Core Award C16420/A18066); and the BRC Genomics Platform at National Institute of Health Research (NIHR) Biomedical Research Centre at Guy's and St. Thomas' Hospitals, London.

## Author contributions

L.A.C., J.S. and S.M.H. wrote the manuscript, performed the experiments and analysed the data; C.G.K., E.C.C., E.H., A.K. and P.L. performed experiments; L.E.L.R. and J.P.C. assisted with confocal experiments and reviewed the manuscript and A.N.A.reviewed the manuscript.

## Competing interests

The authors declare no competing interests.
