## [Peer Review File · Nature Communications]

REVIEWER COMMENTS

Reviewer #1 (T cell memory, CD8 T cells) (Remarks to the Author):

Callender et al. investigate a non-canonical role of the TF GATA3 in regulating CD4+ T cell metabolism and mitochondrial biogenesis during the DDR. GATA3 is generally seen as a regulator of Th2 differentiation, therefore the paper provides a novel role for this TF in regulating functions that go beyond its canonical role. The authors show a previously unknown mechanism of mitochondrial biogenesis in response to DNA damage that is active in highly differentiated CD4+ peripheral T cells. They discover that GATA3 controls a complex of mitochondrial regulators (including PGC1a, NRF2, SOD3 and AMPK) that modulates mitochondrial biogenesis, cell metabolism as well as antioxidant response during DNA damage. Additionally, they show that GATA3 can directly control the expression of mitochondrial genes by binding to DNA.

Major points

1. The main function of GATA3 is to control Th2 specification and to inhibit Th1 differentiation. Rather, the authors focus on stages of differentiation according to the definition of central, effector and terminally differentiated T cells, in line with previous works from the same group. I have no major concerns in this regard, but it is somehow surprising that subsets of memory Th cells are not studied. These could be isolated by combinatorial chemokine receptor expression, such as CCR4+ CCR6- CXCR3- for Th2 cells. Indeed, it would be very interesting to see whether GATA3 can preferentially control mitochondrial biogenesis etc. specifically in this cell subset.
2. Do the authors have whole transcriptomic data or a similar type of global analysis following downregulation of GATA3 in CD4+ T cells? It would be nice to also see the response of these cells following treatment with inducers of DNA damage.

Minor points

1. In Figure 3g, the Authors use naïve T cells for their chromatin immunocleavage experiments. GATA3 is instead overexpressed in memory T cells and terminally differentiated T cells. In this way, the authors could have missed potential interactions at the level of the DNA. Moreover, chromatin accessibility of naïve and memory T cells is largely different, thereby suggesting that additional interactions with the DNA could be present. Why not using total memory or total CD4+ T cells for this assay?
2. Have the authors tried to overexpress GATA3, for instance in activated naïve T cells, and see if it induces a similar phenotype found in memory cells?
3. Most flow cytometry plots/histograms do not have scales. Those should be provided for clarity
4. Figure 2a: TMRE co-staining with antibodies to surface antigens may lead to altered patterns due to antibody signaling (such as anti-CD3). Have the Authors controlled for this possibility?
5. The number of independent experiments is not always specified (e.g., Figure 4k). This should be added.

Reviewer #2 (memory T cells, T helper, plasma cells) (Remarks to the Author):

The authors describe that a transcription factor GATA3 has a new role in the response to DNA damage, influencing mitochondria biogenesis and cell metabolism. The Data of comparison of GATA3 highly and lowly expressed cells and RNA interference of GATA3 support the authors' hypothesis very well. This finding may largely affect the field of cell metabolism, immunology and maybe cancer research. The experiments in the manuscript are well-performed, although there are several minor points to improve.

1. The authors should show a description of the significance (each star) in the first figure legend or the experimental procedures.
2. In the legends of Figure 3d, 4i and ex-4a, the explanations of each gene/protein name (NFE2L2/NRF2) are helpful to readers.

3. Figure 3a can be changed into graphs to be easily understandable.

4. What is AF647 in Figure ex-3a?

5. The authors sometimes utilize Jurkat cells. Please explain the reason or replace data using primary CD4 T cells.

6. The authors use a word 'control' in the title. GATA3 can be a key mediator. However, a function of GATA3 itself during mitochondria biogenesis and cell metabolism is not clear yet. It is not essential to solve the question in this manuscript, but the word 'control' may mislead readers.

Reviewer #3 (GATA3, T cells) (Remarks to the Author):

In this interesting manuscript, Callender et al. investigate the contribution of the GATA-3 transcription factor to mitochondrial biogenesis in human T cells. They find that GATA-3 expression is particularly high in senescent CD45RA⁺CD27⁻ EMRA CD4⁺ T cells. These EMRA CD4⁺ T cells have a high mitochondrial mass and hyperpolarized mitochondria, but maintain normal T cell metabolism. GATA-3 is shown to bind ATR, a protein kinase activated in response to DNA damage, as well as PGC1-alpha, an important regulator for mitochondrial biogenesis, and the antioxidant enzyme SOD3. Next, the authors aim to show that GATA3 can directly regulate mitochondrial biogenesis and fitness. To this end, they use siRNA to reduce GATA3 expression, which resulted in reduced PGC1-alpha protein levels, reduced mitochondrial mass and a shift from hyperpolarized to hypopolarized mitochondrial membrane potentials. The authors postulate that GATA-3 regulates AMPK, which phosphorylates PGC1-alpha. Finally, they conclude that these data provide evidence that GATA3 modulates mitochondrial biogenesis, cell metabolism and antioxidant responses to maintain the viability of cells during DNA damage.

GATA3 is a transcription factor that is crucial for T cell development and function, but its role in the regulation of cellular metabolism is largely unexplored. Therefore, this is an exciting study with high novelty. The effects of GATA-3 on mitochondrial biogenesis are quite convincing. The figures are clear and the manuscript reads well. The main problem that I have with the manuscript is that it remains unclear how GATA-3 exerts its effect.

- For example, GATA-3 may directly regulate PGC1-alpha transcription (given the GATA-3 binding signals in the PPARGC1A locus, Fig. 3d), it may affect the function of the PGC1-alpha protein (GATA-3 and PGC1-alpha form a complex, Fig. 3f) and it may regulate transcription of AMPK, which phosphorylates PGC1-alpha (given the GATA-3 binding signals in the PRKAA1/2 loci, Fig. 3d).

- The authors only show binding of GATA-3 to promoter regions, but whether binding of GATA-3 increases or decreases the expression of AMPK, PGC1-alpha, SOD3, NRF1 and NRF2 is not investigated. These analyses should be done. Certainly, because they show for another gene, c-Myc, that even though GATA-3 is recruited to the Myc promoter region, it has no effect on c-Myc expression.

- The reduction in PGC1alpha expression by GATA-3 siRNA in purified human CD4⁺ T cells is not impressive (MFI decreases from ~1000 to ~850), while the GATA-3 MFI values decrease by ~50%. The finding is difficult to interpret, because the histograms of GATA-3 staining and PGC1alpha staining are very different (compare Ext. Fig. 3B and 3D). GATA-3 expression analysis clearly divides the cell population in a GATA-3-high and a GATA-3-low fraction (Ext. Fig. 3). It is very well possible that PGC1alpha expression is similar in GATA-3-low and GATA-3-high cells.

Minor comments

- All figure legends should have information about the statistics used. I feel that t-tests should not be used, because it cannot be assumed that data have a normal distribution. MFI values in Fig. 1E show large variations, and I'm actually surprised that these values are significantly different between N and EMRA T cells.

- Flow cytometry histograms of PGC1-alpha should be shown (Fig 1E; Fig 4A).

- They authors do not provide evidence for binding of GATA-3 to AMPK, as suggested in Fig. 4J.

- Fig. 4K does not seem to be linked to GATA3 function.

- The discussion is rather superficial and should focus on the mechanisms and molecular

interactions by which GATA3 controls mitochondrial biogenesis (as a transcription regulator or as a binding partner).

Please find below our detailed responses to all reviewers' comments and concerns marked in blue. Underlined italicised text is text that has been added into the revised version of the manuscript.

Reviewer comments

Reviewer 1

Callender et al. investigate a non-canonical role of the TF GATA3 in regulating CD4+ T cell metabolism and mitochondrial biogenesis during the DDR. GATA3 is generally seen as a regulator of Th2 differentiation, therefore the paper provides a novel role for this TF in regulating functions that go beyond its canonical role. The authors show a previously unknown mechanism of mitochondrial biogenesis in response to DNA damage that is active in highly differentiated CD4+ peripheral T cells. They discover that GATA3 controls a complex of mitochondrial regulators (including PGC1a, NRF2, SOD3 and AMPK) that modulates mitochondrial biogenesis, cell metabolism as well as antioxidant response during DNA damage. Additionally, they show that GATA3 can directly control the expression of mitochondrial genes by binding to DNA.

We would like to thank the reviewer for their insightful comments on our manuscript and feel it is much improved following our revisions. Please see below detailed responses to all the comments.

Major points

1. The main function of GATA3 is to control Th2 specification and to inhibit Th1 differentiation. Rather, the authors focus on stages of differentiation according to the definition of central, effector and terminally differentiated T cells, in line with previous works from the same group. I have no major concerns in this regard, but it is somehow surprising that subsets of memory Th cells are not studied. These could be isolated by combinatorial chemokine receptor expression, such as CCR4+ CCR6- CXCR3- for Th2 cells. Indeed, it would very interesting to see whether GATA3 can preferentially control mitochondrial biogenesis etc. specifically in this cell subset.

We have used CCR4, CCR6 along with CXCR3 to define Th subsets in both CD28+ and CD28- CD4 T cell. We find that when DNA damage is induced using hydroxyurea only Th1 subsets increased GATA3 expression and this enhanced expression translated to a rise in mitochondrial mass in the Th1 cells. This was only found in the CD28- T cells with CD28+ showing no effect with the addition of hydroxyurea.

We have added the following data to the results section page 3 and 4:

Indeed GATA3 also increased specifically in CD4⁺CD28⁻ Th1 subsets, defined using CCR4, CCR6 and CXCR3 (Figure 1b and extended data figure 1d, e), following the addition of hydroxyurea, which induces DNA damage via double strand breaks^{13, 14}.

Additionally, Th1 CD4⁺CD28⁻ T cells also displayed an increase in mitochondrial mass following the induction of DNA damage (Figure 1d). Interestingly these changes were only observed in CD4⁺CD28⁻ T cells (Extended figure 1f).

2. Do the authors have whole transcriptomic data or a similar type of global analysis following downregulation or GATA3 in CD4+ T cells? It would be nice to also see the response of these cells following treatment with inducers of DNA damage.

Unfortunately, we do not have any transcriptomic data. We looked to see if we could use publicly available data sets, but none were suitable; therefore, we would need to perform our own analysis however we do not currently have funding to support this.

Minor points

1. In Figure 3g, the Authors use naïve T cells for their chromatin immunocleavage experiments. GATA3 is instead overexpressed in memory T cells and terminally differentiated T cells. In this way, the authors could have missed potential interactions at the level of the DNA. Moreover, chromatin accessibility of naïve and memory T cells is largely different, thereby suggesting that additional interactions with the DNA could be present. Why not using total memory or total CD4+ T cells for this assay?

These experiments were carried out as part of a different project that we were allowed access to in order to validate our findings. We acknowledge that chromatin accessibility is

different between naïve and memory T cells, but the naïve data provided us with good clues as to which pathways GATA3 were involved in, such as mitochondrial biogenesis and energy metabolism.

Furthermore, in light of the article by Ucar et al J Exp Med 2017 214, which suggests that the chromatin changes seen during ageing, both the conversion from naïve to memory as well as chronological ageing, were predominant in the CD8+ T cells. We therefore felt confident to include the data.

2. Have the authors tried to overexpress GATA3, for instance in activated naïve T cells, and see if it induces a similar phenotype found in memory cells?

No this is not something we have done, it's quite challenging to create a stable transduced T cell. We appreciate it is best practice to both overexpress as well as knock-down proteins of interest. However, we don't think that GATA3 knockdown is having a stabilising effect on other proteins, as we observe the loss of downstream proteins which is mirrored when using chemical inhibitors.

3. Most flow cytometry plots/histograms do not have scales. Those should be provided for clarity

We have added scales to all plots.

4. Figure 2a: TMRE co-staining with antibodies to surface antigens may lead to altered patterns due to antibody signaling (such as anti-CD3). Have the Authors controlled for this possibility?

We are aware of this effect; we control for this by always using a TMRE FMO to check that the staining patterns of our subsets does not change with the addition of TMRE. Below is an example of such staining.

We have added the following into the methods section pg12:

Mitochondrial dyes can alter staining patterns, this was checked using fluorescence minus one (FMO).

5. The number of independent experiments is not always specified (e.g., Figure 4k). This should be added.

The number of times we performed each experiment has been made clear in the figure legends.

Reviewer 2

The authors describe that a transcription factor GATA3 has a new role in the response to DNA damage, influencing mitochondria biogenesis and cell metabolism. The Data of comparison of GATA3 highly and lowly expressed cells and RNA interference of GATA3 support the authors' hypothesis very well. This finding may largely affect the field of cell metabolism, immunology and maybe cancer research. The experiments in the manuscript are well-performed, although there are several minor points to improve.

We would like to thank the reviewer for their comments which have helped us to improve the clarity of the paper. Please see below detailed responses to all the comments.

1. The authors should show a description of the significance (each star) in the first figure legend or the experimental procedures.

We have added the significance of each star into the method section as well as including them in every figure legend.

*Stars signify the following: * = $p < 0.05$, ** = $p < 0.01$, *** = $P < 0.005$.*

2. In the legends of Figure 3d, 4i and ex-4a, the explanations of each gene/protein name (NFE2L2/NRF2) are helpful to readers.

We have included both gene and protein names into the legends:

Fig 3(d) Chromatin immunocleavage showing GATA3 binding to the promoter regions of NRF1 (nrf1), NRF2 (nfe2l2), SOD3 (sod3), and PGC1 α (ppargc1a) in naïve CD4⁺ T cells

Fig 4(i) Chromatin immunocleavage showing GATA3 binding to the promoter regions of AMPK α 1 (PRKAA1) and AMPK α 2 (PRKAA2) in naïve CD4⁺ T cells.

3. Figure 3a can be changed into graphs to be easily understandable.

We have changed the graph, plotting γ H2AX against p-p53 and performed a linear regression. The new graph should be easier to interpret as well as highlighting the EMRA subset as having the highest expression of both γ H2AX and p-p53.

4. What is AF647 in Figure ex-3a?

AF647 is the abbreviation for alexa fluor 674, which is a fluorophore which absorbs at 650nm and emits at 665 nm much like allophycocyanin (APC). We have spelt out the abbreviation in the methods section.

5. The authors sometimes utilize Jurkat cells. Please explain the reason or replace data using primary CD4 T cells.

We used both primary CD4 T cells as well as Jurkat T cells owing to the difficulty of working with primary T cells for some experiments. The use of siRNAs together with the induction of DNA damage is challenging due to the need to stimulate the primary CD4 T cells for 2 days prior to electroporation. The induction of DNA damage, together with the siRNA causes increased apoptosis resulting in very few cells and a dramatic change in morphology - even when using the control siRNA. Therefore, for the confocal analysis we used Jurkats, a cell line that is frequently used as a model for studying primary CD4 T cells, this allowed for a shorter incubation with hydroxyurea together with the removal of the pre-stimulation step gave us more robust experimental data.

When we were able to return to the lab owing to social distancing, we were unable to obtain blood from volunteers. The use of leukocyte cones would not do as a substitute as we've found these cells do not perform well for functional assays. Therefore, for a few experiments where we would have liked to use primary CD4 T cells we had to use Jurkats.

Throughout the paper we use primary CD4 T cells to uncover mechanisms and only use Jurkats to validate findings in primary T cells where necessary.

We have been clear to identify which cell type has been used in the manuscript.

6. The authors use a word 'control' in the title. GATA3 can be a key mediator. However, a function of GATA3 itself during mitochondria biogenesis and cell metabolism is not clear yet. It is not essential to solve the question in this manuscript, but the word 'control' may mislead readers.

We have changed the word control to induces. Title now reads:

GATA3 induces mitochondrial biogenesis in primary human CD4+ T cells during DNA damage

Reviewer 3

In this interesting manuscript, Callender et al. investigate the contribution of the GATA-3 transcription factor to mitochondrial biogenesis in human T cells. They find that GATA-3 expression is particularly high in senescent CD45RA+CD27- EMRA CD4+ T cells. These EMRA CD4+ T cells have a high mitochondrial mass and hyperpolarized mitochondria, but maintain normal T cell metabolism. GATA-3 is shown to bind ATR, a protein kinase activated in response to DNA damage, as well as PGC1-alpha, an important regulator for mitochondrial biogenesis, and the antioxidant enzyme SOD3. Next, the authors aim to show that GATA3 can directly regulate mitochondrial biogenesis and fitness. To this end, they use siRNA to reduce GATA3 expression, which resulted in reduced PGC1-alpha protein levels, reduced mitochondrial mass and a shift from hyperpolarized to hypopolarized mitochondrial

membrane potentials. The authors postulate that GATA-3 regulates AMPK, which phosphorylates PGC1-alpha. Finally, the authors conclude that these data provide evidence that GATA3 modulates mitochondrial biogenesis, cell metabolism and antioxidant responses to maintain the viability of cells during DNA damage.

GATA3 is a transcription factor that is crucial for T cell development and function, but its role in the regulation of cellular metabolism is largely unexplored. Therefore, this is an exciting study with high novelty. The effects of GATA-3 on mitochondrial biogenesis are quite convincing. The figures are clear and the manuscript reads well.

We thank the Reviewer for their extensive comments regarding our manuscript and are very pleased that they think our manuscript is novel. We believe their comments have helped us substantiate our conclusions and strengthen our manuscript. Please see below a detailed point-by-point response to all comments:

1. The main problem that I have with the manuscript is that it remains unclear how GATA-3 exerts its effect. GATA-3 may directly regulate PGC1-alpha transcription (given the GATA-3 binding signals in the PPARGC1A locus, Fig. 3d), it may affect the function of the PGC1-alpha protein (GATA-3 and PGC1-alpha form a complex, Fig. 3f) and it may regulate transcription of AMPK, which phosphorylates PGC1-alpha (given the GATA-3 binding signals in the PRKAA1/2 loci, Fig. 3d).
2. The authors only show binding of GATA-3 to promoter regions, but whether binding of GATA-3 increases or decreases the expression of AMPK, PGC1-alpha, SOD3, NRF1 and NRF2 is not investigated. These analyses should be done. Certainly, because they show for another gene, c-Myc, that even though GATA-3 is recruited to the Myc promoter region, it has no effect on c-Myc expression.

We have clarified how GATA3 exerts its effects and would like to address both these comments together.

We had no evidence other than the chromatin immunocleavage that GATA3 had the potential to bind the PRKAA2 (AMPK) gene. We performed an IP using GATA3 and blotted for AMPK and pAMPK but we found no interaction between GATA3 and AMPK despite AMPK being present (and increasing with hydroxyurea) in the loading control. This led us to believe that GATA3 does not regulate AMPK through protein-protein interactions. There is quite a lot of evidence in other cell type for AMPK regulating mitochondrial biogenesis, so we started investigating the idea that AMPK transduced the DNA damage signal to GATA3 and PGC1 α .

First, we definitively showed that pAMPK increased during DNA damage. We then assessed whether DNA damage increased PGC1 α , GATA3 and Nrf2 at the level of transcription or translation. PGC1 α was the only gene to be upregulated by DNA damage. Using an AMPK inhibitor, we showed that the increase in PGC1 α mRNA was largely controlled by AMPK (75% reduction). The expression of GATA3 has been shown to be controlled by enhancing the rate of translation, we tested whether AMPK controlled the amount of GATA3 protein and found that the AMPK inhibitor caused a significant loss of GATA3 protein. We saw no effect on the transcription of Nrf2 following DNA damage we did find increased protein expression. We also investigated whether AMPK caused the translocation of Nrf2 to the nucleus, as shown previously in mouse liver cells. However, we found no change in Nrf2 localisation suggesting that Nrf2 is regulated by the DDR but not AMPK; several publications have indicated that ROS is a strong activator of Nrf2.

These extra experiments have allowed us to put together a much more complete model. We believe the DNA damage response activates the ATM/ATR and AMPK. pAMPK increases expression of PGC1 α at the level of transcription and GATA3 at the level of translation, while the DDR increases Nrf2. These proteins come together with the ATR to form a complex which allow for increased mitochondrial biogenesis.

We have reformulated figure 4 to include this extra data, we have included all representative examples and used inhibitors to prove statements made in the paper. We have also added an extra paragraph into the discussion outlining findings in other cell types.

3. The reduction in PGC1alpha expression by GATA-3 siRNA in purified human CD4+ T cells is not impressive (MFI decreases from ~1000 to ~850), while the GATA-3 MFI values decrease by ~50%. The finding is difficult to interpret, because the histograms of GATA-3 staining and PGC1alpha staining are very different (compare Ext. Fig. 3B and 3D). GATA-3 expression analysis clearly divides the cell population in a GATA-3-high and a GATA-3-low fraction (Ext. Fig. 3). It is very well possible that PGC1alpha expression is similar in GATA-3-low and GATA-3-high cells.

We can explain the slight reduction in PGC1 α expression when using the GATA3 siRNA by our new finding that PGC1 α seems to be controlled predominantly but not entirely by pAMPK during DNA damage. Therefore, while GATA3 does exert some effect on the expression of PGC1 α it was not enough to completely dampen the effect of PGC1 α . Cementing the requirement for AMPK to transduce the DNA damage signal in order to increase mitochondrial biogenesis.

This has been made clear in the text pg 8:

We then assessed gene expression of GATA3, PGC1 α and Nrf2 by qPCR following hydroxyurea treatment in Jurkat T cells. We found only the transcriptional activity of PGC1 α to be upregulated by DNA damage (Figure 4 h). This increase in PGC1 α mRNA was largely controlled by AMPK, for the addition of an AMPK inhibitor, Compound C, significantly reduced mRNA levels (Figure 4i).

We did look at PGC1 α expression in the GATA3-hi and -low populations and while the expression of PGC1 α did seem to be higher in GATA3-hi populations for a few donors, n=3, this was not always the case with the majority of samples showing no difference, n=5.

n=2

n=5

Minor comments

1. All figure legends should have information about the statistics used. I feel that t-tests should not be used, because it cannot be assumed that data have a normal distribution. MFI values in Fig. 1E show large variations, and I'm actually surprised that these values are significantly different between N and EMRA T cells.

We have added statistical information to all figure legends. We do have low n numbers but we did assume normality, we have repeated the analysis using nonparametric tests: Mann-Whitney U test or a Kruskal Wallis test followed by Dunn multiple comparison for post-hoc testing. Both tests are less powerful and for some comparisons they remove significance, we have updated all plots to reflect this. Finally, we believe that some data is significant despite appearances to the contrary as we previously used paired t-tests, subsequently paired Mann-Whitney U tests which are considered more powerful than unpaired t-tests as they use the same participants eliminating variation between the samples that could be caused by anything other than what's being tested.

2. Flow cytometry histograms of PGC1-alpha should be shown (Fig 1E; Fig 4A).

These have been included extended fig 2c and extended fig 4d.

3. They authors do not provide evidence for binding of GATA-3 to AMPK, as suggested in Fig. 4J.

See earlier comment and it is now provided in extended figure 5f.

4. Fig. 4K does not seem to be linked to GATA3 function.

Doxycycline treatment was used to assess the physiological impact of ATR/GATA3/PGC1 α activation. We did try to knock down PGC1 α but it proved lethal to the CD4+ T cells hence the need to use a pharmacological inhibitor. We wished to address the functional consequence of complex activation as without it the increase in mitochondrial biogenesis is meaningless. We show that reducing mitochondrial number in CD4+ T cells caused a dramatic increase in ROS production and a rise in apoptosis, in other words causes the production of cells with a phenotype akin to senescent CD8+ T cell. These senescent CD8+ EMRAs accumulate faster and are more dysfunctional than their CD4 counterparts. It is our belief that this difference is down to the CD4's ability to maintain functional mitochondria.

We have made this clearer in the text pg 9:

as the use of PGC1 α siRNA was lethal to the CD4+ T cells (data not shown).

Doxycycline treatment resulted in a CD4+ T cell that resembled a senescent CD8+ T cell [ref].

Therefore, the presence of GATA3 in CD4+ T cells slows the acquisition of senescence.

5. The discussion is rather superficial and should focus on the mechanisms and molecular interactions by which GATA3 controls mitochondrial biogenesis (as a transcription regulator or as a binding partner).

We have added more about the mechanism and molecular interactions of GATA3 into the discussion.

Mechanistically, GATA3 and AMPK are activated by a DNA damage response, pAMPK increases expression of PGC1 α at the level of transcription and GATA3 at the level of translation. PGC1 α and GATA3 complex together with the ATR and Nrf2 to enhanced mitochondrial biogenesis (Figure 4j).

Evidence for GATA3 being a metabolic regulator is growing, Son et al recently demonstrated GATA3 directly binds to PGC1 α to regulate thermogenesis in white adipose tissue via increased UCP-1 expression³¹. We demonstrate here that in CD4+ T cells PGC1 α is activate both by GATA3 and by AMPK. DNA damage acting through the ATM has been shown to activate AMPK in numerous tissues and can directly interact and phosphorylate PGC-1 α increasing its transcriptional activity³², as seen here. However, we do not find AMPK to regulate the expression of Nrf2 despite it being shown that the translocation of Nrf2 to the nucleus is regulated by the activity of AMPK. AMPK phosphorylates Nrf2 at Ser550, which results in GSK3 β inactivation, both of which are essential for Nrf2 translocation to the nucleus²⁶. However, direct interaction between PGC1 α and Nrf2 has been demonstrated where p38 inactivates GSK3 β , as p38 activity is high in end-stage CD4+ T cells¹² we cannot discount a role for p38 in this model. GATA3 has also been shown to control metabolic reprogramming of CD8+ T cells following stimulation via the T cell receptor, knockdown of GATA3 reduced both glycolysis and oxidative phosphorylation via c-myc³⁴. While we do not find evidence for the involvement of c-myc in our complex it does add weight to GATA3 being a metabolic regulator acting via different mechanisms dependent on cell type and method of activation.

REVIEWER COMMENTS

Reviewer #1 (Remarks to the Author):

The authors have addressed most of my comments. However, they should acknowledge the limitation of having used chromatin data on naive T cells to identify genes bound by GATA3, while GATA3 is preferentially expressed by memory cells. Please also note that my minor comment 4 was referred to the TMRE staining and not to the antibody staining (I apologize if this was not clear), i.e., is the binding of antibodies such as of CD3 changing the distribution of the TMRE staining (this could be very rapid)? Can the authors control for this effect in some way? Thanks.

Reviewer #2 (Remarks to the Author):

Most of my questions were sufficiently answered. Concerning my question to AF647, I wanted the authors to add an explanation which siRNA can be detected by siRNA-labeling AF647, in the figure legend. On the addition, I will recommend publishing the manuscript to the journal.

Koji Tokoyoda

Reviewer #3 (Remarks to the Author):

The authors now provide evidence that activated AMPK increases expression of PGC1 α at the level of transcription and GATA3 at the level of translation. However, it still is not clear how GATA-3 exerts its effect on mitochondrial biogenesis. They do not show whether binding of GATA-3 increases or decreases transcription of the AMPK, PGC1- α , SOD3, NRF1 and NRF2 genes. On the contrary, they now conclude that the increase in PGC1 α upon DNA damage was largely controlled by AMPK and that PGC1- α expression in the GATA3-hi and GATA3-low populations are not different. The authors claim that they have clarified how GATA3 exerts its effects but show then that AMPK controlled the amount of GATA3 protein, which is a different issue.

Please find below our detailed responses to all reviewers' comments and concerns marked in blue. Underlined italicised text is text that has been added into the revised version of the manuscript.

Reviewer comments

Reviewer 1

The authors have addressed most of my comments. However, they should acknowledge the limitation of having used chromatin data on naïve T cells to identify genes bound by GATA3, while GATA3 is preferentially expressed by memory cells. Please also note that my minor comment 4 was referred to the TMRE staining and not to the antibody staining (I apologize if this was not clear), i.e., is the binding of antibodies such as of CD3 changing the distribution of the TMRE staining (this could be very rapid)? Can the authors control for this effect in some way? Thanks.

1. We have acknowledged the limitations as follows:

We next sought to identify other mitochondrial regulators that may also be upregulated by GATA3 using chromatin immunocleavage on resting naïve CD4 T cells. Despite GATA3 being preferentially expressed by memory cells, these data revealed that

2. Apologies for the confusion. Changes in Ca²⁺ concentration can mediate mitochondrial membrane permeabilization, papers have shown that transient changes in Ca²⁺ concentration have no effect on mitochondrial membrane potential, however Ca²⁺ overload can trigger mitochondrial depolarization (Roy & Hajnoczky. Methods 2008 46(3) 213). Antibody binding can elicit changes in Ca²⁺ levels but the effect is transient. We stain for TMRE in complete media and our RMPI contains HEPES, capable of buffering Ca²⁺. We assumed that as all our samples had been treated the same way (solutions, antibodies, timing) that any observed difference was due either to ER stress that occurs with differentiation and results in a sustained release of Ca²⁺ or the sample treatment. We have used FCCP to interfere with the proton gradient in both unstimulated and stimulated CD4⁺ T cells, which reduced TMRE staining – see below.

Reviewer 2

Most of my questions were sufficiently answered. Concerning my question to AF647, I wanted the authors to add an explanation which siRNA can be detected by siRNA-labeling AF647, in the figure legend. On the addition, I will recommend publishing the manuscript to the journal.

1. We have altered the figure legends to reflect this.

GATA3 siRNA-AF647

Reviewer 3

The authors now provide evidence that activated AMPK increases expression of PGC1 α at the level of transcription and GATA3 at the level of translation.

However, it still is not clear how GATA-3 exerts its effect on mitochondrial biogenesis. They do not show whether binding of GATA-3 increases or decreases transcription of the AMPK, PGC1-alpha, SOD3, NRF1 and NRF2 genes. On the contrary, they now conclude that the increase in PGC1 α upon DNA damage was largely controlled by AMPK and that PGC1-alpha expression in the GATA3-hi and GATA3-low populations are not different.

The authors claim that they have clarified how GATA3 exerts its effects but show then that AMPK controlled the amount of GATA3 protein, which is a different issue.

We would like to thank the reviewer for their comments which have helped to further improve the clarity of the paper.

We have taken their comments on board and performed qPCR with and without GATA3 siRNA under conditions of DNA damage. This data is now included in extended data figure 5b. Interestingly, we find that following GATA3 knock down the expression of PGC1a and AMPK mRNA increase and SOD3, NRF1 and NRF2 decrease. An increased PGC1a transcription despite a downstream signaling impairment has been reported previously in a model of impaired metabolic regulation (Aksentijevic et al. *Frontiers in Physiology* 2020). Our data suggests that we are observing an energetic stress response caused by the impairment of metabolic and redox control. It is well reported that an increase in ROS generation activates PGC1a and AMPK, switching their roles from energy regulation to ROS protection in order to prevent apoptotic cell death associated with altered mitochondrial function.

We conclude that GATA3 controls the transcription of SOD3 and NRF1 and NRF2. We have shown that GATA3 plays an important role in metabolic regulation.

The additional work performed to address Reviewer's comment has been described as follows:

To determine whether GATA3 changed the transcription of the AMPK, PGC1 α , SOD3, NRF1 and NRF2 genes we performed further siRNA knockdowns in the presence of hydroxyurea (Extended data figure 5b). Paradoxically knockdown of GATA3 caused increased expression of AMPK and PGC1 α mRNA and a reduction in the transcription of SOD3 and NRF1 and NRF2. Increased PGC1 α transcription despite downstream signaling impairment has been observed previously and was associated with an energetic stress response²⁴. Our data would suggest that GATA3 controls the transcription of key redox sensing components and that its loss leads to metabolic stress²⁵.

REVIEWERS' COMMENTS

Reviewer #1 (Remarks to the Author):

Accept. Congratulations
Enrico Lugli

Reviewer #3 (Remarks to the Author):

The authors have sufficiently addressed my remaining concerns